# Proteome Discoverer—A Community Enhanced Data Processing Suite for Protein Informatics

**DOI:** 10.3390/proteomes9010015

**Published:** 2021-03-23

**Authors:** Benjamin C. Orsburn

**Affiliations:** Department of Pharmacology and Molecular Sciences, Johns Hopkins University, Baltimore, MD 21205, USA; borsbur1@jhu.edu

**Keywords:** proteomics, protein informatics, glycoproteomics, mass spectrometry

## Abstract

Proteomics researchers today face an interesting challenge: how to choose among the dozens of data processing and analysis pipelines available for converting tandem mass spectrometry files to protein identifications. Due to the dominance of Orbitrap technology in proteomics in recent history, many researchers have defaulted to the vendor software Proteome Discoverer. Over the fourteen years since the initial release of the software, it has evolved in parallel with the increasingly complex demands faced by proteomics researchers. Today, Proteome Discoverer exists in two distinct forms with both powerful commercial versions and fully functional free versions in use in many labs today. Throughout the 11 main versions released to date, a central theme of the software has always been the ability to easily view and verify the spectra from which identifications are made. This ability is, even today, a key differentiator from other data analysis solutions. In this review I will attempt to summarize the history and evolution of Proteome Discoverer from its first launch to the versions in use today.

## 1. Introduction

Today, there are over 1000 different tools for the processing of proteomics data [1]. This daunting number counts both active software and tools that have fallen by the wayside during the rapid evolutions toward the realization of analyzing all the proteins in a biological system. The primary search tools used by most researchers today are a much smaller number. While any surveys that have been conducted have their own biases, today most users are likely divided primarily between MaxQuant [2,3], Mascot [4], SpectroNaut [5], Proteome Discoverer and PEAKS [6] with the rest of users divided between other powerful commercial and open source solutions that are often linked to a user’s history or background. For example, many new trainees in proteomics that are migrating from genomics or transcriptomics gravitate toward the use of R-based proteomics pipelines [7,8]. In addition, in western Europe near where they are currently developed, you will find a larger number of users of OpenMS [9] and CompOmics [10,11] pipelines. 

Proteome Discoverer (PD) is a commercial product of Thermo Fisher Scientific that was first released in 2007 as a replacement for the aging BioWorks proteomics framework. PD has evolved over the years from humble beginnings as little more than a wrapper for Sequest and Mascot. Today it is a powerful and flexible environment with both commercial and freely available tools to address nearly any proteomics workflow, from both labeled and label-free quantification to crosslink analysis, glycoproteomics and even top– down proteomics [12,13]. PD has always had limitations and critics have rightfully pointed out that relying on a commercial software team for development and deployment with mandatory rounds of internal troubleshooting will always produce a software package that is behind the times at launch [14]. However, for users who appreciate a steady and tested interface and a help line for software issues, PD has been a stalwart companion of an increasing number of researchers in this field. In this text, I will attempt to detail the history of PD as well as highlight the key points in the development of the PD environment, as well as what sets it apart from other tools in use at the time and today. This list is in no way meant to be comprehensive, and in the lack of references please assume that these statements are derived from lessons learned as an operator of every version of this software released to date.

## 2. Common Themes

Over the last 14 years and 11 main releases of PD, the software has evolved to meet the challenges faced by proteomics scientists at that point in time. Figure 1 is a summary of some of the key changes in the evolution of PD. However, four common themes have been present in all versions of the software.

***Flexible data input:*** While PD is, at heart, a product of Thermo Fisher Scientific and intended for the processing of instrument files from that vendor, PD has always accepted data from other vendor platforms when the files have been converted to universal formats. Today, PD can accept any mass spectrometry data files that have been converted to Mascot Generic Format **[15]**, mzML **[16]** and mzXML **[17]**. Furthermore, data from previous iterations of Thermo software including BioWorks and previous versions of PD, as well as output files formatted in the mzData **[18]** format can be imported into PD for filtering and analysis.

***The combination of search results:*** An early characteristic that set PD apart from other solutions at the time was the ability to combine results from multiple search engines. PD 1.0 released with a direct interface to on-drive or network linked instances of Mascot as well as on-drive installations of Sequest and the ZCore search engine. ZCore was an engine specifically designed for the searching of MS/MS spectra that were the result of electron transfer dissociation [19]. Today, results from seven or more distinct search engines can be combined into a single output. In addition, multiple instances of the same search engine can be utilized in series or in parallel for complex experiments. Perhaps the best example of this flexibility was exhibited in Rinas et al., where 11 separate search engines were utilized for the fast photochemical oxidation of proteins (FPOP), with settings optimized to reflect the biochemical likelihood of the oxidative modifications occurring on each amino acid [20,21].

***Direct access to the unaltered MS/MS spectra leading to each identification****:* Perhaps the most striking central philosophy of PD compared to other search tools has been the constant ease of access to the original MS/MS data. For comparison, the popular and powerful MaxQuant data analysis package had no direct way to visualize the MS/MS spectra from which identifications were made until 2015 [22]. Popular data processing pipelines such as MSFragger [23] rely on secondary programs such as the Proteome Data Viewer for the visualization of peptide spectral match data [24]. MSFragger does have direct access to spectral data, but only when MSFragger is operated within the PD environment. 

As software tools in proteomics continue to improve, particularly with the addition of more intelligently designed false discovery rate (FDR) estimation and filtering tools, it is tempting to trust identifications without verifying the spectra themselves. However, many of the most embarrassing retractions that have occurred in proteomics could have been prevented by the verification of the peptide spectral matches (PSMs) by trained mass spectrometrists [25,26,27]. For a more comprehensive discussion on the manual analysis of PSMs, please see the thoughtful review and tutorial by Dr. Simone König titled “spectral quality overrides software score” [28]. By placing the unaltered MS/MS spectra within easy reach at all times in its output interface, PD helps to ensure the validity of identifications made by automated tools. A screenshot of the PD 1.0 interface displaying this unchanged core philosophy of the program is shown in Figure 2. 

***Rapid porting from discovery to validation:*** The final key thread throughout all versions of PD is the ability to migrate identifications made within the software rapidly to methods for targeted validation. In the earliest versions of PD, peptides and proteins identified could be directly exported to the now retired PinPoint software from the same vendor. Today, results can be directly exported into an inclusion list format for all Orbitrap instruments. In addition, filtered proteins and peptides of interest can be exported in various formats, including the direct generation of FASTA files from these filtered proteins and peptides. Most powerfully, however, the PD output files can be directly imported into a number of tools for validation and analysis thanks to collaborative efforts between the PD development team and those at Proteome Software, [29] OptysTech [30] and the Skyline software of the University of Washington. [31,32]

## 3. Software Architecture and Data Formats

All versions of PD appear to be compiled in Microsoft Visual Studio with all core framework components based on NET frameworks in the C# programming language. Accepted input formats have increased over time, with today’s iteration accepting all versions of the Thermo RAW instrument outputs as well as flexible input of variations of other universal mass spectrometry file formats including Mascot Generic Format (MGF), mzML, mzXML and mzData [15,16,17,18]. Sequencing information for matching against MS/MS spectra can be accepted by default with the accurate parsing of all standard protein FASTA files. A flexible parsing system with definable rules was added in later versions to facilitate the input of non-traditional formats such as the files from “next-generation” sequencing instruments. Default data output in all versions up to 1.4 are a MSF SQLite database file that can be readily opened by generic SQLite database tools as well as more specialized proteomics tools such as Scaffold, M2Lite and MS2Go [33]. PD versions 2.0 and above produce both an MSF file as well as a second SQLite pdresult file. In these versions, the MSF contains the peptide spectral match data and the pdresult file containing the assembled and filtered values used to construct the peptide and protein identifications. For downstream validation, targeted LCMS tools such Pinpoint, Pinnacle and Skyline can directly import MSF and pdresult files and use these processed results to create targeted mass spectrometry methods. Processed data in nearly every version can be exported directly to Microsoft Excel format as well as to CSV, with recent versions providing the option for header formats compatible with the increasingly popular statistical programming language R.

## 4. A Brief History of Proteome Discoverer Versions and Key Highlights

**PD 1**.**0**, **2007:** The initial launch version of PD, v1.0 was more of a skeleton of the framework that it is today, but multiple central components remain unchanged. PD 1.0 was compatible with Windows XP 32-bit, in line with most commercial computer hardware of the time. PD 1.0 featured the Sequest and ZCore search engines with direct compatibility with Mascot input and output. The output of all three engines, if applicable, could be combined into a single output file and the peptide spectral matches could be directly visualized for each peptide and protein. PD 1.0 was compatible with all Thermo RAW files and could accept files in Mascot Generic Format (MGF). PD 1.0 output was, as it is today, an SQLite database file. Data processing occurred primarily using “wizards” that walked the end user step-wise through a logical data processing workflow.

**PD 1**.**1**, **2009:** The release of PD 1.1 in 2009 saw a larger migration of users from the popular BioWorks software from Thermo due to the relative ease of the interface compared to its predecessor and promotions that allowed users of BioWorks to freely transition to the new software interface. To obtain high coverage proteomes on any instrument at this point in history, extensive two dimensional fractionation was a critical step in the workflow, either offline or online with then-popular methods such as MuDPiT [34]. The support for fractionated samples was added in this version. The commercial release of PD 1.1 added the now familiar node-based logical workflow interface. PD 1.1 also saw improvements in the universal input and output nature of the software with import capabilities extended to mzXML and outputs of mZData. This release of the software was also accompanied with the Daemon executable, which allowed a central computer to function as the main data processing center and up to 20 PCs to send data to that PC for processing. In addition, the Daemon could be configured through Xcalibur processing methods to enable the automatic transfer, queuing, and processing of acquired LCMS files. These events could be programmed to occur either in parallel with the queue acquisition or immediately following the completion of a sample queue for fractionated samples. These functions, though rarely employed, still exist in every version today.

**PD 1**.**2**, **2010:** The 2010 release of PD added many more key functions in popular use in proteomics at the time, particularly stable isotope labeling and the first addition of false discovery rate filtering in the software. Stable isotope labeling by amino acids in cell culture (SILAC) had, by this time, been labeled as the “gold standard in proteomics quantification” and variations on the laborious process of labeling all the proteins in cells in culture were the focus of most proteomic research [35,36]. Similar techniques, such as dimethyl labeling, were utilized for samples that could not be grown in culture for the days necessary to fully incorporate these expensive amino acids [37,38]. PD 1.2 added new nodes for quantification from the MS1 spectra to support multiple isotopic labeling techniques. In addition, a new precursor ion area detector node was added that allowed label-free quantification based on the abundance of the three most intense peptides from each identified protein. The addition of these critical key workflows led an even larger base of customers to migrate to the PD interface from BioWorks and MaxQuant. The addition of false discovery rate (FDR) estimation and filtering by target decoy [39,40] was a major step toward improving the validity of peptide and protein identifications by the software. A final key improvement in PD 1.2 was the addition of a separate installation file for the Viewer program, which could be installed on any desktop computer and allowed multiple users to transfer processed MSF files and evaluate results simultaneously. The Viewer allowed end users to both visualize the peptide spectral matches and to filter processed data locally without reprocessing.

**PD 1**.**3**, **2011:** The next release of PD featured a landmark event for the further development of the software: the integration of software developed by external groups. This came in the form of the first Percolator node and phosphoRS. The use of Percolator for FDR estimation and filtering is now commonplace, with various iterations of the semi-supervised PSM re-evaluation tool in use in nearly every proteomics software today [41]. Improvements on the base Percolator design have often focused on simply increasing the speed [42] of this pressure-tested solution or in applying it to alternative datasets [43]. Although PD was compatible with 64-bit architecture, it functioned in emulation mode, as all key components of the software, including Percolator, were natively 32-bit. The second external tool for PD was the first of many nodes produced by the Institute of Molecular Pathology (IMP). PhosphoRS is similar to the aScore algorithm and uses secondary evidence in the PSM to provide a localization confidence score for the possible sites of phosphorylation within that peptide [44]. Further node development occurred at the IMP, including the development of a new search engine specifically designed for high resolution MS/MS spectra, MSAmanda, as well as multiple accessory nodes [45]. The IMP-MS2 spectrum processor node allowed for the deconvolution of high resolution MS/MS spectra to produce a new spectra where all baseline resolved fragment ions could be replaced with their corresponding single charged fragment ions. The removal of obvious isotopes could also be simultaneously performed. Later iterations of this node allowed for the reassignment of the monoisotopic mass in the case of an inaccurate assignment of an isotope as the monoisotopic ion by the instrument.

The IMP-Spectrum Merger node addressed the common instrument method design of the time in reporter ion quantification, where two MS2 scans were obtained for each parent ion. In this workflow, collisional-induced dissociation (CID) scans in the ion trap were used for peptide sequence identification. A second fragmentation event using higher energy collisional dissociation (HCD) of the parent ion was used to liberate the reporter ion fragments for quantification in the Orbitrap. Two scan events were utilized due to technical limitations of both the early Orbitrap hardware and the reagents of the time. Early iterations of the LTQ Orbitrap instruments had difficulty controlling HCD fragmentation and ion transfer to the mass analyzer. The Orbitrap XL featured multiple versions of the HCD cell throughout its lifetime in order to mitigate these challenges [46]. The most popular isobaric reagent of the time, the iTRAQ 8-plex reagent, featured chemistry that required significantly higher collisional energy to fully liberate the reporter region than was optimal for peptide identification [47]. The dual fragmentation method helped alleviate both limitations and allowed for higher multiplexed quantification, albeit at slower acquisition rates. The IMP-Spectrum Merger allowed for the combination of the data from these two scans into a single MS/MS spectrum.

The IMP-Post match search recalibrator allowed the results of a search to be used in tandem with the spectrum exporter node to readjust all monoisotopic masses in the exported file. The exported file could then be searched with tighter mass tolerances at the precursor level to obtain results with higher confidence. While many of these tools were compatible with PD 1.3, these were primarily used in house at IMP and associated institutions. These nodes became accessible to the wider community with the launch of pd-nodes.org, which corresponded more closely to the release of the next version of PD.

**PD 1**.**4**, **2013:** From 2005 to 2018, each release of Orbitrap instrumentation arrived with an increase in the number of fragmentation spectra that could be obtained in the same amount of time [48,49]. The corresponding increase in overall data density gradually added additional pressure to existing tools and pipelines. In 2011, the first Orbitrap analyzer with a smaller internal diameter and corresponding increase in curvature of the electric field [50] was released in the form of the LTQ Orbitrap Elite system. The reduction in Orbitrap size in addition to an enhanced Fourier transform algorithm resulted in a 2× increase in the overall data density. While this giant step in hardware performance played a key role in the completion of high profile proteomic studies such as the NCI-60 proteome project [51,52] and the first two drafts of the human proteome [53,54], this massive increase in data density pointed out multiple weaknesses in proteomic informatics of the day. Updates in PD 1.4 helped alleviate many of these hurdles with native 64-bit compatibility and the first instance of the Sequest algorithm that was capable of multithreading. In addition, PD 1.4 brought the first use of spectral library search nodes in the form of the SpectraST [55,56] engine, originally developed by the Institute of Systems Biology and the MSPepSearch node long used as a stand-alone tool available from the US National Institute of Standards [57]. On the commercial front, newcomer ProteinMetrics enabled the direct use of both Preview and Byonic through nodes in PD 1.4 for labs with licenses for these packages and local installation [58]. After launch, a faster version of Percolator was released for customers struggling with long processing times and this version would be available at install in the next iteration of the software.

**PD 2**.**0**, **2014:** The basic framework of PD 1.0 was designed to meet the demands of proteomic scientists in 2007. As the field developed through the decade, experimental designs became increasingly complex due to more ambitious overall study goals. The PD 1.0 framework required a comprehensive overhaul to address this increase in complexity and to meet the new demands of the time. No studies more aptly encapsulated the potential and challenges of proteomic informatics of the day than the two human proteome drafts released in *Nature* in 2014. In order to compile the data from highly fractionated samples from multiple organs in PD, the fractions from each organ would realistically need to be processed separately and combined with external tools [53]. Reanalysis of these studies identified other flaws in the PD architecture with the identification of false discoveries in the initial analysis that were biologically improbable [59]. PD 2.0 launched with a new experimental design interface and the separation of workflows into a processing step to create MSF files containing PSMs and a consensus workflow for assembling the PSMs into peptides and proteins. The expansion into two parts required several new nodes, the most notable perhaps being the critical addition of FDR tools to estimate errors at the peptide and protein level, in addition to the traditional tools for PSM filtering.

Further community support for the PD environment arrived in the form of two new nodes from the OpenMS community. The first, the LFQProfiler, brought powerful MS1 feature-based quantification to PD. The second, RNPxl, enabled the identification of peptides that were bound to RNA fragments [60]. In addition, new nodes were available for purchase that enabled, for the first time, top-down proteomic analysis in PD in the form of nodes that could directly connect PD with the ProsightPC software tools [61].

**PD 2**.**1**, **2015:** PD 2.1 featured primarily improvements to the general structure of PD 2.0 with multiple tweaks in the study design and a series of bugfixes inevitable in the release of an entirely new software architecture.

One subtle change for researchers came in the form of a new strategy for protein group inference. In all previous iterations of PD, when identified peptides could be assigned equally to multiple proteins in the FASTA file being examined, the protein with the highest percent coverage was chosen for display. A consequence of this strategy was that the shortest protein isoforms and protein fragments were displayed more commonly than the full length sequence. In PD 2.1 this strategy was flipped and the protein with the longest sequence was chosen as the “master” protein when equal evidence exists to support two proteins or proteoforms.

The most powerful improvements to the PD 2.1 interface would come from external developers in the form of the IMP-PD nodes which, for the first time, allowed for the full use of the PD interface without any purchases. The IMP-PD nodes effectively replaced any nodes that the vendor paid royalties on by replacing these steps in the pipeline with open-source alternatives. IMP-PD features other tools and capabilities beyond the commercial released version with downstream quantitative analysis through the LIMMA node, which executed statistical analysis through a local iteration of R.

A common and accurate criticism of protein informatics is the lack of complex statistical models in use in other -omics fields [62]. To address these shortcomings, PD 2.1 included more quantitative statistical tools than any previous iteration with intuitive filtering through volcano plots and principal component analysis data reduction tools. IMP-PD improved on this further through the first two nodes that directly interfaced with the R statistical programming language. IMP-apQuant (originally PeakJuggler) and IMP Normalization and LIMMA brought powerful tools for quantitative analysis from tools in R directly into the PD interface [63].

IMP-PD further extended the open source capabilities of the software with the introduction of the glycoproteomics search tool SugarQb. SugarQb uses an iterative search functionality whereby spectra possessing oxonium fragment ions characteristic of glycopeptide fragmentation are searched separately using a large list of glycan moieties [64]. SugarQb currently installs with a glycan modification library of approximately 1600 mammalian glycan chains. Glycopeptide analysis has long been possible using the commercial Byonic program, but SugarQb opens the growing field of glycoproteomics to a wider number of scientists.

Two other accessory programs appeared for this version, MS2Go [65], for the production of output reports with information most biologists are actively seeking. Finally, a useful stand-alone tool from IMP, the PD node manager can be indispensable for power PCs loaded with multiple versions of PD and even some of the second party nodes described in this text. The PD node manager provides instant insight details of the tools and versions installed on that local computer.

**PD 2**.**2**, **2017:** The next release of PD brought a dramatic improvement in label-free quantification in PD, in the form of the Minora nodes. All previous versions of PD utilized a precursor ion node in which the intensity of the most abundant parent ions from each protein were compared between runs. False discoveries due to excessive shifts in the retention time were filtered by the end user and no retention time alignment algorithm was in use. Retention time alignment was, at that time, a central component of multiple tools including MaxQuant, IonStar, Progenesis and OpenMS. [9,66]

Additional new features in PD came from external development in the world of crosslinking analysis. The XlinkX software brought powerful new tools to a field increasingly focused on the direct analysis of protein–protein interactions. New crosslinking reagents and intelligent crosslink identification capabilities made possible in the Orbitrap Tribrid architecture ushered in a new era in protein crosslinking studies by enabling, for the first time, some progress toward the dream of true in vivo crosslinking studies [67,68,69].

An additional major external update was the release of the MSFragger-PD nodes. MSFragger is one example of the recent next generation proteomics search tools possessing true open search functionality. The MSFragger PD nodes allows for the identification of unknown post-translational modifications within the PD environment. [23] The MSFragger PD nodes also bring in the PeptideProphet FDR tools which can be used in conjunction with other PD nodes. [70] It is also notable that MSFragger primarily exists in its original form and through the FragPipe environment as Java-based tools. The direct implementation of this with the PD environment is a demonstration that other Java tools may be made compatible with PD.

**PD 2**.**3**, **2018:** The next commercial PD release was highlighted by drastic improvements in the XlinkX nodes and a new functionality that allowed the visualization and filtering of the synchronous precursor selection (SPS) steps in MS3-based isobaric quantification experiments. Through post-processing analysis, peptides could be discarded when evidence was present to suggest that the SPS functioned inadequately in picking fragment ions derived from the parent for quantification [71]. Improved functionality allowed for direct links between protein reports and pathway analysis web tools, including KEGG, WikiPathways and Reactome [72,73,74].

**PD 2**.**4**, **2019:** The ASMS 2019 release of PD brought native support of chimeric spectra through SequestHT and a flurry of new powers for developers in the shape of open and flexible scripting nodes. Examples of the application of these nodes at ASMS primarily centered on interfacing with R, although there is no reason that other programs and batch scripts cannot be executed through the node. A central focus for the proteomics field should continue to be on the improvement and advancement of statistical models. The integration of tools already existing in R, including the powerful MSStats package, could be an ideal next step toward meeting these goals [75,76,77]. Figure 3 is a screenshot from PD 2.4, demonstrating some of the capabilities of the software that have been described, including the use of the MSAmanda 2.0 engine, IMP-Hyperplex and visualization functions for quantification between samples.

**PD 2**.**5**, **2020:** The most obvious changes in the newest release of PD come from internal developments from the vendor. To meet an increased external interest in the use of deep learning tools for proteomics, the vendor has licensed new tools named Inferys. Inferys utilizes CPU-based deep learning, similar to the Prosit GPU-based algorithms [77]. While limited in the number of cores in Central Processing Unit architecture, compared to the thousands of processing cores in modern Graphics Processing Units, Inferys is a valuable new addition to the PD environment. Inferys can be utilized in multiple ways in PD, for the construction of deep learning-based spectral libraries as well as for rescuing MS/MS spectra that were incorrectly discarded as low quality by other tools in the PD pipeline. Currently, Inferys is only compatible with unlabeled peptides that have been reduced and alkylated with iodoacetamide and digested with trypsin. The PD 2.5 manual indicates that further developments in this toolkit are underway for later versions.

## 5. Conclusions: A Biphasic Future for Proteome Discoverer?

Since the initial release of PD, upgrades of the software have often been either free or accompanied with marginal costs through “maintenance” license agreements. PD is going in two clear directions, with powerful open-source tools continuing to grow at a steady rate in parallel with commercial tools, beginning with PD 2.5 upgrades to the newest versions with licensed content require upgrade fees to cover royalty fees incurred by the vendor. It is easy to see a biphasic future of PD, with both open and licensed versions beginning to further diverge. A fully licensed commercial version of PD with all available accessory licenses, including ProsightPD and Byonic, carried a total price tag of nearly USD 30,000 when initially purchased by our facility. Nearly a dozen operators have processed data for scores of their own projects over the years, reducing the cost to pennies per file searched. At this time, there is no reason to not install the second party nodes described in this text in the commercial version and these are increasingly used by our core operators, often in conjunction with the commercial tools. In the opposite phase, students increasingly have fully functional free versions of PD with various combinations of the nodes described in this text that are the most applicable to their own research goals.

I use other software in my research and will continue to do so, but when a proteomics tool identifies an important peptide or the presence of a post-translational modification of interest, I always seem to default to reanalysis with tools in PD to get me to the original spectra from which that identification was made. For 14 years, PD has been the fastest way to find that spectra and to verify that trained human operators agree with that identification, and it is hard to see that changing anytime soon.

## Figures and Tables

**Figure 1 proteomes-09-00015-f001:**
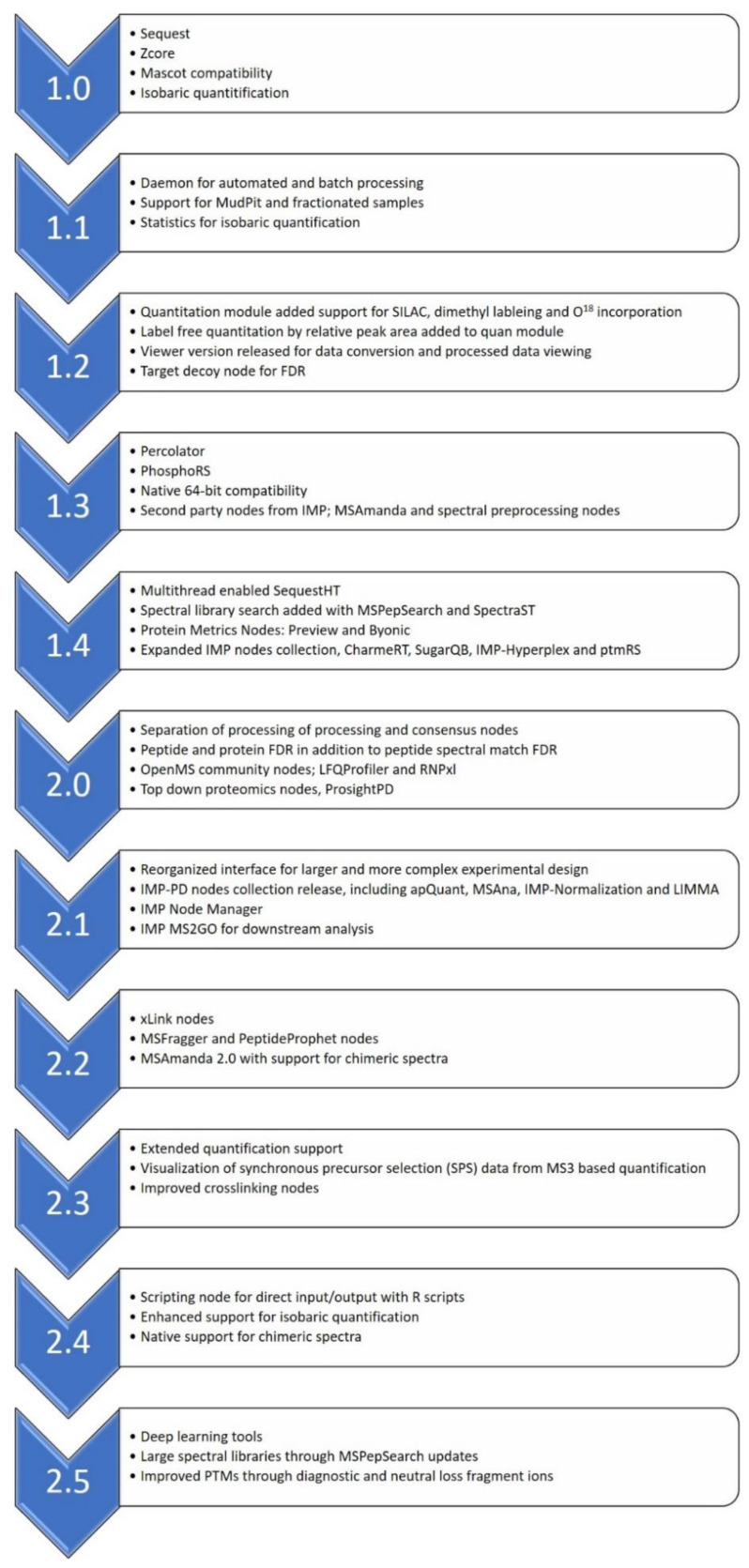
A summary of PD versions and key highlights of each version release. FDR: false discovery rate.

**Figure 2 proteomes-09-00015-f002:**
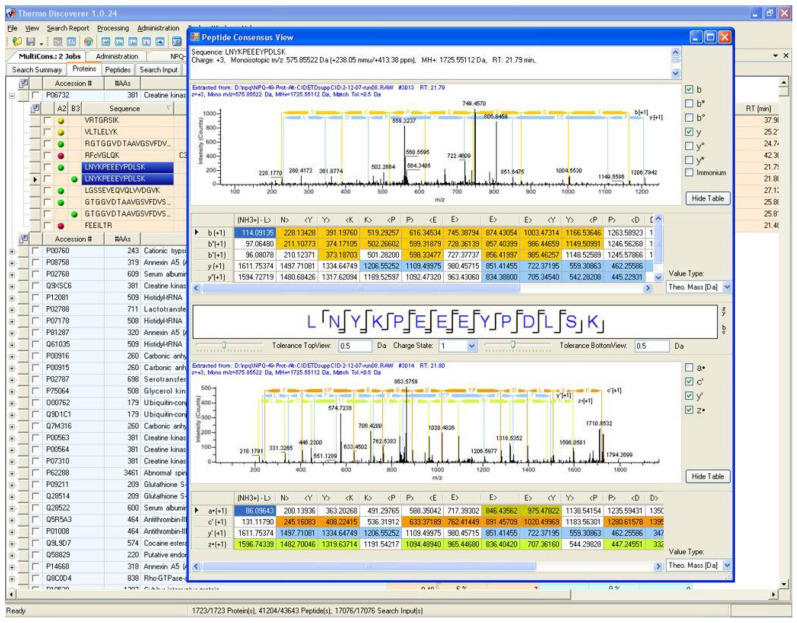
A result screenshot from the first release of Proteome Discoverer (PD) in 2007. In this output protein identification, supporting PSMs leading toward that identification and the original MS/MS spectra can be easily visualized and verified for quality by the end user.

**Figure 3 proteomes-09-00015-f003:**
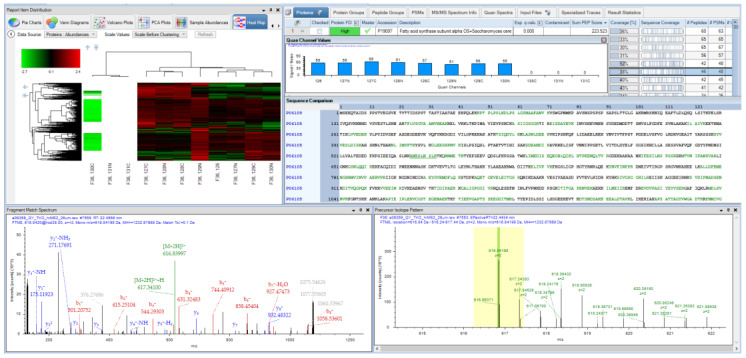
A result screenshot from PD 2.4 operated entirely with open source community developed nodes, demonstrating both searching and quantification functions.

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
