# Peer review of "Proteome Discoverer—A Community Enhanced Data Processing Suite for Protein Informatics"

_proteomes, 2021, doi:10.3390/proteomes9010015_

Round 1

Reviewer 1 Report

This is a comprehensive review of the development of Proteome Discoverer, an important software for computational proteomics analysis. The author is the right person to write such a review due to his history of using, reporting on, educating, and helping researchers with the Proteome Discoverer issues and questions. The main strength of the manuscript is the comprehensive description of the chronological developments that took place between the first and the latest version of the software. Such a manuscript would be of interest not only to seasonal proteomics researchers but also to newcomers as they will be able to follow recent, and not so, developments in computation proteomics and how they affected the software.

I think that the manuscript is a unique and valuable contribution and should be considered for publication after minor revision.

Major comments:

  • The title doesn't match the main body of information. The title of the manuscript is very specific about what the manuscript should deliver. More specifically, "central focus on the visual verification". There is however too little content dedicated to the unique characteristics that enable that said verification and also an explanation or example as to why is that a central focus. I don't consider the content of Figure 1 and the more personal take on peptide spectrum match verification, at the end of the manuscript as enough to justify the title of the manuscript. For the main text to match the title, more should be said about the peptide-spectrum match visualization capabilities. Alternatively, the biphasic nature and future of the software can be the main point of the title as a lot of detail has gone into describing this aspect.

Minor comments:

  • lines 28-29, please include a mention of and a reference for Spectronaut. This software is in many aspects as advanced as Proteome Discoverer with the main difference that it is dedicated to the analysis of data-independent acquisition data.
  • lines 79-80, the statement, "MaxQuant data analysis package has no direct way to visualize the MS/MS spectra from which identifications were made" is incorrect. The majority of the latest MaxQuant versions have included an integrated viewer where one can interactively pull out and examine individual peptide-spectrum matches. 
  • The content of Figure 1 is too outdated.  When looking at the peptide-spectrum match visualization in Proteome Discoverer 2.4, I find many differences compared to the one shown in Figure 1. In order to strengthen the point of the software offering powerful visualization, I would suggest that a more up to date example is included.

Author Response

Thank you for the very helpful suggestions! 

I have:

1) Changed the title of the manuscript thanks to your suggestion

2) Clarified my statements regarding MaxQuant and applied the reference for the MaxQuant viewer addition to the software in 2015

3) Added SpectroNaut and a reference

4) I've added an additional figure (3) to the manuscript to demonstrate the capabilities of PD 2.4 though the use of a free version utilizing exclusivley community provided nodes for identification and quantification. 

Reviewer 2 Report

The author of the Review is undoubtedly a specialist in the use of mass spectrometry for proteomic research and the MS/MS data processing. The paper describes the evolution of Proteome Discoverer (Thermofisher) - two versions of the software - commercial and free, and provides a history of successive versions of the software. 

The aim of Proteomes Review manuscript  should  provide concise and precise updates on the latest progress made in a given area of research.

In my opinion, the manuscript does not meet this condition.  I  do not understand what is the aim of the work: history of the software or description of advantages of the latest version.   The manuscript seems to be an advertising material with product placement.  No justification why PD is better than many other software except that possibility of usage of any data set in FASTA format. What is interesting, the author writes that he himself uses other software (which one? why? ) from time to time he verifies data with PD.

Author Response

Thank you for reading and reviewing this work and I apologize that this appeared to be an advertisement and for not being more clear regarding the logic for why I wanted to write this.

A quick literature search I performed this morning identified over 100 published papers in 2021 where Proteome Discoverer was used as the search tool. I found examples of the use of every version from 1.3 through 2.4. I thought it would be helpful to have a review out there that could explain what the differences are between versions. As an example, I added some new text regarding how protein inference is performed differently between versions 1.0 through 2.0 that was altered in PD 2.1 (section PD 2.1).  Some other alterations were applied to the text and a change of title was performed to meet other reviewer suggestions. 

Round 2

Reviewer 2 Report

The author of this publication is clearly an expert in the use of PD software for proteomic data analysis. Perhaps his article will be useful especially to users of older versions of the software. The changes made to the text are minor and my assessment of the scientific value has not changed. 

Author Response

Over 600 papers have been published in 2021 that have used Proteome Discoverer and this is would be the first peer reviewed paper that describes what it is and how it processes data.